# Performance Evaluation for Repair of HSGc-C5 Carcinoma Cell Using Geant4-DNA

**DOI:** 10.3390/cancers13236046

**Published:** 2021-11-30

**Authors:** Dousatsu Sakata, Masao Suzuki, Ryoichi Hirayama, Yasushi Abe, Masayuki Muramatsu, Shinji Sato, Oleg Belov, Ioanna Kyriakou, Dimitris Emfietzoglou, Susanna Guatelli, Sebastien Incerti, Taku Inaniwa

**Affiliations:** 1Department of Accelerator and Medical Physics, Institute for Quantum Medical Science, QST, Chiba 263-8555, Japan; abey@riken.jp (Y.A.); muramatsu.masayuki@qst.go.jp (M.M.); sato.shinji@qst.go.jp (S.S.); inaniwa.taku@qst.go.jp (T.I.); 2Department of Charged Particle Therapy Research, Institute for Quantum Medical Science, QST, Chiba 263-8555, Japan; suzuki.masao@qst.go.jp (M.S.); hirayama.ryoichi@qst.go.jp (R.H.); 3Veksler and Baldin Laboratory of High Energy Physics, Joint Institute for Nuclear Research, 141980 Dubna, Russia; dem@jinr.ru; 4Institute of System Analysis and Management, Dubna State University, 141980 Dubna, Russia; 5Medical Physics Laboratory, Medical School, University of Ioannina, 45110 Ioannina, Greece; ikyriak@uoi.gr (I.K.); demfietz@uoi.gr (D.E.); 6Centre For Medical Radiation Physics, University of Wollongong, Wollongong 2522, Australia; susanna@uow.edu.au; 7Centre d’Études Nucléaires de Bordeaux Gradignan, CNRS/IN2P3, UMR5797, Université de Bordeaux, F-33170 Gradignan, France; incerti@cenbg.in2p3.fr

**Keywords:** Geant4-DNA, DNA repair, cell surviving fraction

## Abstract

**Simple Summary:**

To evaluate the repair performance of HSGc-C5 carcinoma cell against radiation-induced DNA damage, a Geant4-DNA application for radiobiological research was extended by using newly measured experimental data acquired in this study. Concerning fast- and slow-DNA rejoining, the two-lesion kinetics (TLK) model parameters were adequately optimized (the repair speeds of each process were reasonably close to the DNA rejoining speed of the nonhomologous end-joining and homologous recombination pathways). The lethality probabilities of the DNA damage induced by complex double strand breaks (DSBs) and binary repair were approximately 3% and 40%, respectively. Using the optimized repair parameters, the Geant4-DNA simulation was able to predict the cell surviving fraction (SF) and the DNA repair kinetics.

**Abstract:**

Track-structure Monte Carlo simulations are useful tools to evaluate initial DNA damage induced by irradiation. In the previous study, we have developed a Gean4-DNA-based application to estimate the cell surviving fraction of V79 cells after irradiation, bridging the gap between the initial DNA damage and the DNA rejoining kinetics by means of the two-lesion kinetics (TLK) model. However, since the DNA repair performance depends on cell line, the same model parameters cannot be used for different cell lines. Thus, we extended the Geant4-DNA application with a TLK model for the evaluation of DNA damage repair performance in HSGc-C5 carcinoma cells which are typically used for evaluating proton/carbon radiation treatment effects. For this evaluation, we also performed experimental measurements for cell surviving fractions and DNA rejoining kinetics of the HSGc-C5 cells irradiated by 70 MeV protons at the cyclotron facility at the National Institutes for Quantum and Radiological Science and Technology (QST). Concerning fast- and slow-DNA rejoining, the TLK model parameters were adequately optimized with the simulated initial DNA damage. The optimized DNA rejoining speeds were reasonably agreed with the experimental DNA rejoining speeds. Using the optimized TLK model, the Geant4-DNA simulation is now able to predict cell survival and DNA-rejoining kinetics for HSGc-C5 cells.

## 1. Introduction

Radiation treatment is one of the most widely used therapeutic techniques for cancer treatment aiming at depriving tumor cells of their reproductive potential [1]. The investigation of biological responses to radiation, such as reproductive cell death, has emerged as a multiscale and multidisciplinary area of research interest. As a trigger, radiation-induced crucial DNA damage represented by double-strand breaks (DSBs) can be a cause of reproductive cell death [1,2]. However, because of the experimental requirements, it is difficult to directly observe and quantify the details of such microscopic lesions. Due to this difficulty, indirect measurements of features related to initial DNA damage, such as physical disconnection of DNA fiber [3,4], chromosomal aberrations [5], and phosphorylated H2AX as a marker for DSBs [6,7,8], have been attempted to further investigate radiation-induced DNA damage. Thus, track-structure Monte Carlo (MC) simulations have an important role in investigating DNA damage induced after radiation irradiation [9,10,11]. In the past decades, many MC codes have achieved successful outcomes for quantitative investigation of radiation-induced initial DNA damage within cellular domains and subcellular biological components [12,13,14,15,16,17,18,19,20,21,22,23,24,25,26,27,28,29].

In a previous study [30], Geant4-DNA simulations [31,32,33,34], an extension of Geant4 [35,36,37] for low-energy particle transport, including the simulation of water radiolysis and geometries of biological targets, successfully estimated not only initial DNA damage, but also the fraction of surviving V79 cells by using a two-lesion kinetics (TLK) model [38,39] that bridges the gap between initial DNA damage and reproductive cell death. Although V79 is one of the most important cell lines in radiation biology research field, it is important to extend the application to other cell lines used for the evaluation of radiation treatment effects to provide a more comprehensive and robust simulation framework. In general treatment systems, clinically delivered doses to a target tumor are calculated from the so-called “biological dose,” which is the absorbed physical dose multiplied by the relative biological effectiveness (RBE) representing the cell-killing effectiveness of the irradiation [40,41]. RBE tends to be evaluated as the ratio of the cell survival of the reference cell line to the cell survival of cells irradiated with gamma rays [42,43]. A type of human cancer cell, HSGc-C5, is widely used for evaluating RBE for proton/carbon treatment planning [44,45,46,47,48,49]. For this reason, we investigated extending the Geant4-DNA application with a TLK model for performance evaluation of the DNA damage repair mechanism in the HSGc-C5 cell line.

## 2. Materials and Methods

This study is comprised of two parts: The first part is an experimental study (illustrated in Section 2.1) to measure cell surviving fractions (SFs) and DNA rejoining kinetics as the reference data for the optimization of the TLK model parameters of HSGc-C5. The second part describes the optimization of the TLK model parameters to evaluate the repair performance of HSGc-C5 (illustrated in Section 2.2).

### 2.1. Experimental Condition

#### 2.1.1. Irradiation Condition

All experiments were performed in the cyclotron facility at the National Institutes for Quantum and Radiological Science and Technology (QST, Chiba, Japan). The protons were delivered in the experimental beamline (C8) from the cyclotron (NIRS-930; Thomson CSF, La Défense, France). A Wobbler method was used to widely spread the field of the primary protons [50] to deliver a uniform beam field. The energy of the protons was 70 MeV, and the collimated field size was approximately 8 cm × 8 cm at the isocenter plane. The primary energy of the protons upon the cell entrance was changed between binary choice by inserting a 32 mm thick poly methyl methacrylate (PMMA) block (just before the Bragg peak of a 70 MeV proton in the PMMA block as discussed in Section 3.1).

##### Dosimetry

The beam-monitor count (counts/Gy) and the beam-count rate (counts/second) were calibrated by using a Markus ion chamber (PTW 23343; PTW, Freiburg, Germany) as shown in panel (A) of Figure 1, and the calibration was repeated for both conditions, with and without the 32-mm thick PMMA block. In the biological assay, the cultured cells were plated on the back side of the upstream window of plastic cell-culture flasks (Falcon 353107/353108; Corning Inc., Corning, NY, USA), as shown in panels (B) and (C) of Figure 1. Therefore, the water-equivalent thicknesses of the chamber window and of the flask window were different, and in order to minimize the water-equivalent thickness difference 0.41 mm thick PMMA sheet was placed in front of the chamber.

In the biological simulation study that followed, the energy spectra of protons at the cell entrance level were estimated. For this purpose, the material properties of PMMA (mass density and mean ionization potential, the so-called I-value) needs to be estimated. Thus, we have also measured the depth dose for 70 MeV protons in PMMA by changing the PMMA thickness ≤35 mm as the reference data for the estimation of the material properties with the dosimetry setup. The dose measurements were repeated five times at each depth.

##### Colony Assay

Panel (B) of Figure 1 illustrates the schematic experimental setup for the colony assay. A T-25 plastic cell-culture flask was used to place the cells in the irradiation field. A plastic fixing jig was used downstream to fix the flask on the beam line. The target cells were plated on the back side of the window located at the isocenter. To measure cell survival for two different radiation qualities, the irradiation experiments were performed with and without a 32 mm PMMA block. The delivered dose was chosen as 1, 2, 3, 5, or 7 Gy for the irradiation condition without a PMMA block, and as 1, 2, 3, 4, or 6 Gy for the irradiation condition with a PMMA block. The dose rate was approximately 1 Gy/min for all conditions. The irradiation experiments for each condition were repeated three times on the same day, but the experiments with and without a 32 mm PMMA block were performed on different days.

##### Gel Electrophoresis Assay

Panel (C) of Figure 1 illustrates the schematic experimental setup for electrophoresis assay. The target cells are confined in a T-12.5 flask placed at the isocenter. To inhibit DNA rejoining during irradiation and prevent the need to taxi them to the biological work room the flask was cooled on ice. Precooling on ice was also performed from 5 min before the irradiation. To the cell sample, 200 Gy was delivered at a dose rate of approximately 100 Gy/min.

#### 2.1.2. Cell Culture and Biological Processing before Irradiation

We selected a cell line that is a typical benchmark cell in radiation therapy, the HSGc-C5 cell line (No. JCRB1070), distributed by the Japanese Collection of Research Bioresources (JCRB) Cell Bank (National Institutes of Biomedical Innovation, Health, and Nutrition, Japan). The cell line was stored at the NIRS as frozen stocks after culturing in a 5% CO${}_{2}$ incubator at 37$\;{}^{\circ }C$ within Eagle’s minimum essential medium containing 60 mg/L kanamycin, supplemented with 10% fetal bovine serum (Equitech-Bio Inc., Kerrville, TX, USA). As the first step, approximately 10 days before the irradiation, the frozen stocked cells were unfrozen by placing them in warm water at 37$\;{}^{\circ }C$, and then inoculated and sub-cultured into T-75 flasks (Falcon 353135; Corning Inc., Corning, NY, USA). Two days before the irradiation, the sub-cultured cells were trypsinized (2.5% Trypsin, Thermo Fisher Scientific, Waltham, MA, USA) and inoculated into the T-25 (for the colony assay, Falcon 353018; Corning Inc., Corning, NY, USA) or T-12.5 (for the fraction of activity released (FAR) assay, Falcon 353107; Corning Inc., Corning, NY, USA) flask. The cellular densities in both types of flask were kept at 3.24 $\times \;10^{4}$ cells/cm${}^{2}$ as a confluent condition.

#### 2.1.3. Cell Survival Measurement

Within 40 minutes after irradiation, the irradiated cells were taxied to the work room, rinsed with phosphate-buffered saline (PBS), and trypsinized. The experimental procedure performed in this study was described previously in reports by Suzuki [51,52]. Then, the trypsinized cells were plated onto plastic culture dishes (Falcon 353002; Corning Inc., Corning, NY, USA). After the cells were incubated in the CO${}_{2}$ incubator at 37$\;{}^{\circ }C$ for approximately 14 days, the colonies were fixed and stained with 20% methanol containing 0.2% crystal violet. Any colony consisting of more than ≥50 cells was scored as a surviving clone. The SF was calculated according to the following equation:(1)$$\mathrm{SF}=\frac{N_{\mathrm{irr}}^{\mathrm{col}}/N_{\mathrm{irr}}^{\mathrm{plat}}}{N_{\mathrm{nonirr}}^{\mathrm{col}}/N_{\mathrm{nonirr}}^{\mathrm{plat}}},$$
where $N_{\mathrm{irr}}^{\mathrm{col}}$ is the number of irradiated cells that create a colony, $N_{\mathrm{irr}}^{\mathrm{plat}}$ is the number of cells plated onto the dish after irradiation, $N_{\mathrm{nonirr}}^{\mathrm{col}}$ is the number nonirradiated cells (usually called as control cells) that create a colony, and $N_{\mathrm{nonirr}}^{\mathrm{plat}}$ is the number of control cells plated onto the dish.

#### 2.1.4. DNA Rejoining Kinetics Measurement

The cell sample was kept at 4$\;{}^{\circ }C$ during exposures. As for the colony assay, the irradiated cell samples were taxied to the work room within 40 min after irradiation but kept on ice to maintain a cold temperature. The experimental procedure performed in this study was described previously in a report by Hirayama [53]. The cells were lysed directly or kept in the incubator to allow DNA rejoining under the aerobic conditions with 5% CO${}_{2}$ at 37$\;{}^{\circ }C$. The cells were washed twice with cold PBS, rinsed with cold 0.05% trypsin-EDTA, and kept on ice for 20 min. The cells were resuspended in cold PBS and embedded in 1% SeaPlaque GTG agarose gels (50111; Cambrex, Baltimore, MD, USA) plugs at a density of approximately 1 × 10${}^{5}$ cells/mL (1 × 10${}^{4}$ cells/plug). All steps were performed on ice to minimize DNA rejoining. The cells in the agarose plugs were incubated in a lysis solution (R&D Systems, Minneapolis, MN, USA) containing 0.5 mg/mL proteinase K (Sigma-Aldrich, Saint Louis, MO, USA) for 1 h to guarantee the diffusion of chemicals into the agarose. Cell lysis was performed at 50$\;{}^{\circ }C$ for 24 h. The plugs were equilibrated at a pH of 8 in tris-ethylenediaminetetraacetic acid (TE) buffer (Sigma-Aldrich, Saint Louis, MO, USA) for 1 h at room temperature and used for electrophoresis. The plugs were loaded onto 0.6% SeaKem Gold agarose gels (50152; Lonza, Switzerland) and subjected to electrophoresis at a field strength of 0.6 V/cm in 0.5× Tris-borate EDTA (TBE) buffer (GeneMax, Taiwan) for 36 h. The gel was stained for ≥3 h with ethidium bromide (2 μg/mL) and maintained overnight at room temperature in distilled water. The fluorescence intensities were measured with a UV transilluminator (Mupid-Scope WD; Mupid, Japan) and a digital camera (IXY 220F; Canon, Japan) with an orange filter, which was connected to a computer with an image analysis software (1D Image Analysis Software; Kodak, Japan).

The fluorescence intensities of DNA that was retained in the plug and released from the plug were measured by using the image analysis software. The fluorescence intensities for released DNA were proportional to the total amount of DNA fragments, which were separated by physical disconnection of the DNA fiber. The fraction of activity released (FAR) calculation equation was as follows:(2)$$\mathrm{FAR}\left(t\right)=\frac{I_{\mathrm{out}}\left(t\right)}{\left(I_{\mathrm{in}}\left(t\right)+I_{\mathrm{out}}\left(t\right)\right)},$$
where, *t* is the time after irradiation, $I_{\mathrm{in}}\left(t\right)$ is the fluorescence intensity of DNA retained in the plug, and $I_{\mathrm{out}}\left(t\right)$ is the fluorescence intensity of DNA released from the plug. Then, the relative FAR referenced to FAR (0 min) as a function of time was calculated.

### 2.2. Simulation Conditions and Model Calculation

#### 2.2.1. Calculation of the Incident Proton Energy Spectra at the Cell Entrance

To estimate the initial DNA damage by means of Geant4-DNA, the energy of the incident protons at the cell entrance was determined using Geant4 (since Geant4-DNA does not support particle transport in PMMA, and the spatial resolution of Geant4 was sufficient to simulate particle transport in millimeter scale volumes). In this study, the energy of the incident protons was downscaled by filtering through a 32 mm PMMA block. Hence, the incident energy of protons was not mono-energetic but was multienergetic due to the energy loss and straggling in the PMMA block. The energy loss distributions depend upon n-times inelastic interaction obeying the Poisson distribution and the cross sections of the energy loss for each interaction. In addition, even when the PMMA block was not placed upstream, the proton energy spectrum was broadened when the protons passed through the window of the plastic flask. MC simulation is a useful tool for estimating the energy spectra at the cell entrance if the properties of the materials, such as mass density, atomic/molecular composition, and mean ionization potential (so-called I-value), are known when simulations are performed using the condensed-history approach.

However, in general, it is hard to know the precise material properties of organic materials, such as PMMA. In this study, we have adjusted the material properties, in particular, the I-value, in a way that was consistent with the measured depth dose in the PMMA block, by comparing with the Geant4 simulations. PMMA is a composite material consisting of five carbon atoms, eight hydrogen atoms, and two oxygen atoms. The mass density was selected as the typical PMMA density of 1.190 g/cm${}^{3}$. The absorbed dose in the sensitive volume of an advanced Markus chamber was simulated for a PMMA block and a PMMA sheet, with PMMA block thicknesses ≤35 mm to adjust the I-value. For simulations of particle transport in the PMMA block/sheet and in the dosimeter, QGSP_BIC_EMY was chosen as the condensed-history particle transport model. QGSP_BIC_EMY is the physics set combining a standard electromagnetic model (G4EmStandard_option3) and a binary cascade hadronic model (G4BinaryCascade). This is the recommended particle transport model set for protons in the clinical energy range [54,55]. During the simulations with the adjusted PMMA properties, the energy of protons at the entrance to the sensitive volume of the advanced Markus chamber is measured.

#### 2.2.2. Initial DNA Damage

##### Simulation Using Geant4-DNA

To estimate the initial DNA damage, the same simulation configuration as in the previous study was used [24]. A geometrical model of cell that imitates a normal human fibroblast cell was used (bottom panel of Figure 1), where a cell nucleus (14.2μm×14.2μm×5.0μm) was placed at the center of a water absorber modeling the cytoplasm (28.0μm×28.0μm×5.0μm). In the former, a subbiological component was assembled with the total number of base pairs (bps) being approximately 6.4 Gbp. A double-helix DNA fiber consists of spherical phosphate/deoxyribose molecules with two ellipsoidal nucleotide bases (the combination of the pair was chosen randomly) as a backbone [56,57] constructed by forming a fractal shape chromatin fiber wrapped by spherical histones [23,24]. 56,400 incident protons for the 0 mm PMMA block configuration and 11,400 incident protons for the 32 mm PMMA block configuration were homogeneously irradiated on the top of the cell nucleus, 3.0μm away from the center plane to the other side of the cell, as shown by black arrows in the figure. The energy of the protons was randomly chosen from those in the estimated proton energy spectrum for each PMMA thickness illustrated in Section 3.1. The average proton energies were approximately 68.5, 18.7, and 10.8 MeV, and the standard deviations of the spectra were 0.5, 1.7, and 2.1 MeV at PMMA thicknesses of 0, 30, and 32 mm, respectively. The corresponding unrestricted linear energy transfer (LET${}_{\infty }$) values were 0.05, 0.60 and 0.96 keV/μm [58], respectively.

For particle transport in cell and reactions with cellular subcomponents, the G4EmD- NAPhysics_option4 set of physics models, was used below 10 keV and G4EmDNAPhysic-s_option2 was used above 10 keV (up to 1 MeV) [59,60,61]. The production and reaction schemes of chemical species during radiolysis were simulated using the independent-reaction time (IRT) method [62,63]. The direct and indirect DNA damage models were adjusted in the previous study [24]. The adjustment in the proportional probability direct damage model that the probability of direct damage increases proportionally from 0 at 5 eV to 1 at 37.5 eV was selected as originally proposed by PARTRAC [17].

As a result of the adjustment in the previous study [24], we set 0.405 as the probability of a chemical reaction between a hydroxyl radical and the sugar-phosphate backbone resulting in an indirect damage. The histones placed in the cell model are assumed to be the perfect scavengers for all radiolytic species which leads to a 5% DSB yield reduction [24].

##### DNA Damage Classification

The initial DNA damage needs to be classified into two components because the TLK model considers two types of repair kinetics. As an assumption, we considered that all simple DSBs are repaired by the fast-repair process, and all complex DSBs are repaired by the slow-repair process. In this study, as in the previous study, a classification scheme originally proposed by Nikjoo et al. [14] for DSB damage classification was used. Simple DSBs can be considered to be two-strand breaks (SB) on opposite strands within a short distance (typically within 10 bps) from each other. We considered two damage types as complex DSBs: DSB${}_{+}$ requires a DSB and at least one additional SB within 10 bp, whereas a DSB${}_{++}$ requires at least any two DSBs DSB/DSB${}_{+}$) along the chromatin fiber segment. Each damage cluster is defined as different from another if no damage can be found in 100 consecutive bps.

#### 2.2.3. Evaluation of Repair Performance

##### TLK Model

The TLK model proposed by Stewart [38] represents the kinetic processes of fast- and slow- DNA repair as well as the subsequent SF calculated in accordance with their residual lethal DNA damage. The number of lesions induced by radiation increases during the irradiation, then such lesions repaired in the following DNA repair processes over time. Both the fast- and slow- repair consider simple rejoining of bp-break ends at the same position as expressed as L(t) at time *t*, but the corresponding repair process is different. Whereas, multiple-lesion repairs (second-order repair) consider two bp-break ends at the different position expressed as L(t)L(t) which may result in a complex aberration, possibly by incorrect rejoining of the break ends with two different lesions (the wrong pair rejoining: binary misrepair).

In this study, the six parameters TLK model was applied using the same approach as used in the previous studies with V79 cells [30,64] as well as the original study by Stewart [38]. The model parameters can be categorized into two types: (1) repair probability coefficients, which represent the fraction of rejoined lesions in a unit of time (λ or η), and (2) parameters for lethality which represent the probability of the residual lesion leading to cell death (β or γ). With these considerations, the six parameters TLK model can be written as follows,
(3)$$\frac{dL_{1}\left(t\right)}{dt}=D\left(t\right)Y\Sigma_{1}-\lambda_{1}L_{1}\left(t\right)-\eta L_{1}\left[L_{1}\left(t\right)+L_{2}\left(t\right)\right],$$
(4)$$\frac{dL_{2}\left(t\right)}{dt}=D\left(t\right)Y\Sigma_{2}-\lambda_{2}L_{2}\left(t\right)-\eta L_{2}\left[L_{1}\left(t\right)+L_{2}\left(t\right)\right],$$
and
(5)$$\frac{dL_{f}\left(t\right)}{dt}=\beta_{1}\lambda_{1}L_{1}\left(t\right)+\beta_{2}\lambda_{2}L_{2}\left(t\right)+\gamma \eta {\left[L_{1}\left(t\right)+L_{2}\left(t\right)\right]}^{2}.$$

Here $L_{1}\left(t\right)$ ($L_{2}\left(t\right)$) is the number of lesions in fast- (slow-) repair per cell at a time *t* from the start of irradiation. $L_{f}\left(t\right)$ is the number of lethal lesions leading to cell death at *t*. $D\left(t\right)Y\Sigma_{1}$ and $D\left(t\right)Y\Sigma_{2}$ are the lesion production terms for fast- and slow- lesions, respectively, which are proportional to the dose rate D(t) multiplied by the instantaneous lesions in a unit dose rate per unit number of bps Σ (Gy${}^{-1}$ Gbp${}^{-1}$) and number of bps in a cell *Y* (Gbp).

In this study, we assumed that all simple DSBs underwent the fast-repair process, and that all complex DSBs underwent the slow-repair process. Thus, the instantaneous lesions are defined as $\Sigma_{1}=N_{DSB}$ and $\Sigma_{2}=N_{DSB+}+2N_{DSB++}$, where the N${}_{DSB}$, N${}_{DSB_{+}}$ and N${}_{DSB_{++}}$ are the number of simple DSBs, DSB${}_{+}$, and DSB${}_{++}$, respectively, as in Nikjoo’s definition. $\lambda_{1}$, $\lambda_{2}$ and η are the rates of rejoined lesions (h${}^{-1}$) by the fast-, slow-, and binary-rejoining processes, respectively. Similarly, $\beta_{1}$, $\beta_{2}$, and γ represent the probabilities of the residual lesions leading to cell death for each rejoining process. As in the previous studies [30,38], $\beta_{1}$ was forced to 0, since in general, simple DSBs do not have much of an effect on cell survival ≥2 weeks after irradiation. If the first-order repair is not saturated, the half-life τ of the rejoining can be calculated by τ=ln2/λ.

Finally, these yields were numerically integrated to calculate SF
(6)$$\mathrm{SF}\left(t\right)=\ln\left(-L_{f}\left(t\right)\right)=\ln\left(-\int_{0}^{t}\left(\beta_{1}\lambda_{1}L_{1}\left(t\right)+\beta_{2}\lambda_{2}L_{2}\left(t\right)+\gamma \eta {\left[L_{1}\left(t\right)+L_{2}\left(t\right)\right]}^{2}\right)dt\right).$$

The differential equation has been solved numerically by means of the fourth order Runge-Kutta method in the boost/numerical C++ library. The SF is calculated at t=336 h since the number of colonies is counted after 14 days from the irradiation in the experimental assay. Additionally, D(t) is set to 60 Gy/h until the target dose is delivered. The time step of the integration is set to $1\times 10^{-4}$ h.

##### Random-Breakage Model

FAR is a method for quantitative investigation of the number of fragments separated by physical disconnection of the DNA fiber, such as DSBs.

According to the random-breakage model [65,66,67], FAR can be calculated from the number of the unrejoined DSBs ($(L_{1}\left(t\right)+L_{2}\left(t\right))/Y$) using the following equation:(7)$$\mathrm{FAR}\left(t\right)=F_{max}\left\{1-\left[1+K\left(L_{1}\left(t\right)+L_{2}\left(t\right)\right)/Y\left(1-\frac{K}{M_{0}}\right)\right]\exp^{-K(L_{1}\left(t\right)+L_{2}\left(t\right))/Y}\right\},$$
where $F_{max}$ is the maximum fraction of the DNA that can enter the gel plug, $M_{0}$ is the average DNA length in a chromosome, and *K* is the detection limit length (DNA fragments shorter than *K* do not move out of the gel wall). In this study, $F_{max}$ was set to 1, $M_{0}∼139$ Mbp (6.4 Gbp/46 chromosomes) and K=1Mbp is the limit of the FAR assay (measured with the fiducial DNA marker). To match the experimental definition, to calculate the relative FAR, the FAR values were scaled by applying FAR($t_{0}$), where $t_{0}$ is the time when the irradiation was stopped.

##### Parameter Optimization

To evaluate the repair performances represented as $\lambda_{1}$, $\lambda_{2}$, η, $\beta_{1}$, $\beta_{2}$ and γ, these model parameters in Equations (3)–(5) were optimized for HSGc-C5, in a way that was consistent with the experimentally measured SF and relative FAR. To optimize the parameters, we use the Ceres Solver [68], which is an open-source C++ library based on the nonlinear least-squares method for solving optimization problems. The key computational cost is the solution of a linear least squares problem of the form for parameters *x*:(8)$$\arg\min_{\Delta x}\frac{1}{2}{\left|J\left(x\right)\Delta x+F\left(x\right)\right|}^{2},$$
where F(x) is a matrix of *n*-dimensional vector of variable (the number of data points), and *m*-dimensional function of *x* (the number of optimized parameters). J(x) is the Jacobian. In this study, SPARSE_NORMAL_CHOLESKY was selected as the algorithm of the Jacobian factorization, since the problem is sparse in general. The residual cost for each data point was calculated as $V_{calc}-V_{exp}$, where $V_{calc}$ was calculated as the value with simulated DSBs and $V_{exp}$ was calculated as the value of experimental data with the same weight for all configurations of both the SF and relative FAR. Then, the problem can be simply considered as the problem of finding the minimum of the function,
(9)$$\min\frac{1}{2}\sum_{i}^{n}{\left|V_{calc,i}-V_{exp,i}\right|}^{2},$$
as a function of residual cost of each data point (represented as *i*-th).

## 3. Results

### 3.1. Incident Energy Spectra at the Cell Entrance

To estimate the energy spectra of the incident protons at the cell entrance in the irradiation experiments for the colony assay and FAR assay, the I-value of the PMMA used in the experiments was evaluated to be 65 eV to provide the best agreement with the measured depth dose as the I-value. Figure 2 shows the relative dose distribution of 70 MeV protons scaled at 0 mm. The maximum standard errors of the experimental data are 0.27% before the Bragg peak and 8% after the peak. The Bragg peak occurred between 32 mm and 33 mm. Thus, in this study, we selected a PMMA thickness of 32 mm to downscale the energy of the incident protons.

Figure 3 shows the energy spectra of the incident protons at the entrance of the cell, scaled to the maximum of the number of protons at a 0 mm PMMA thickness. The energy is shifted to lower values with increasing PMMA thickness reaching 10.8 MeV at 32 mm, and the width of a spectrum broadens due to the energy losses and energy straggling while passing through the PMMA. Even when the PMMA block is not inserted, the protons will lose their energy when passing through the PMMA sheet and the window of the advanced Markus chamber. The average proton energies were approximately 68.5, and 10.8 MeV, and the standard deviations of the spectra were 0.5 and 2.1 MeV at PMMA thicknesses of 0 and 32 mm, respectively. The corresponding unrestricted linear energy transfer (LET${}_{\infty }$) values were 0.05, 0.60, and 0.96 keV/μm [58], respectively.

### 3.2. Initial DNA Damage

The simulated initial number of DSBs for each damage type is shown in Figure 4. For all damage types, the numbers of DSBs were slightly larger with a 32 mm PMMA block than the numbers without a PMMA block. The average numbers of DSBs were 4.11 ± 0.14 and 0.74 ± 0.11 Gy${}^{-1}$Gbp${}^{-1}$ for simple-DSB and complex-DSB without a PMMA block, respectively. With a PMMA block, the average numbers of DSBs were 4.69 ± 0.17 and 1.04 ± 0.12 Gy${}^{-1}$Gbp${}^{-1}$, respectively.

### 3.3. Optimized Repair Performance

As shown in Figure 5, the model parameters of the TLK model are reasonably optimized for the Geant4-DNA to reproduce both SF and relative FAR.

The optimized parameters are presented in Table 1. Through the fast-repair process, the probability of the repair was approximately 3.36 h${}^{-1}$ (the half-life time is approximately 12.6 min). Through the slow-repair process, the probability of the repair was approximately 0.01 h${}^{-1}$ (the half-life time is approximately 70.0 h). The probability of the binary repair was significantly small ($4.58\times 10^{-6}$ h${}^{-1}$) relative to that of single rejoining. In contrast, the lethality of binary repair was very high (probability of repair leading cell death, possibly misrepair, in the binary repair process ∼40%). Compared with the lethality of binary repair, the lethality of residual complex DSBs was relatively small (∼3%).

## 4. Discussion

The simulated DSBs are increasing with decreasing the incident particle energy (increasing with LET). The simulated yields of initial DSB are very similar to the DSB yields simulated with mono-energetic protons in the low-LET domain of the previous study (5 to 6 Gy${}^{-1}$ Gbp${}^{-1}$) [23]. The yield ratio of DSB${}_{+}$/DSB${}_{++}$ is approximately 1.44 at 0 mm PMMA block, while the yield ratio approximately 1.13 at 32 mm PMMA block. Since the number of DSB is significantly larger than the number of DSB${}_{+}$, the number of type change from DSB to DSB${}_{+}$ is larger than the number of type change from DSB${}_{+}$ to DSB${}_{++}$ when the LET is getting larger in this LET range. As shown in Figure 4, the differences among the number of DSBs for each damage type appear to be small but this is not surprising, when considering LET value at this energy (LET${}_{\infty }$ = 0.96 keV/μm at 10.8 MeV [58]). However, considering only complex DSBs, the number of DSBs was increased by 43% when a PMMA block was inserted. When a 32 mm PMMA block was inserted and the PMMA thickness was close to the Bragg peak position, the average energy of the protons was approximately 10.8 MeV (both the CSDA range and projected range of 10.8 MeV protons in PMMA were approximately 0.52 mm [58]). This fact also means that it is hard to downscale the energy of protons more using PMMA blocks at 1 mm step. Hence, to perform the experiments for higher-LET protons, we need to perform the experiments with thinner PMMA blocks for lower energy than 10.8 MeV, otherwise we need to use low energy proton beam facilities.

In general, cells rely on two highly regulated DSB repair pathways: the nonhomologous end-joining (NHEJ) pathway and homologous recombination (HR) pathway. Our results indicate that NHEJ and HR regulate the DNA repair process of the HSGc-C5. This was more evident when we compared the optimized speed of the rejoining with the measured repair speed. According to the experimental measurements, NHEJ processes can be completed in approximately 0.5 hour, whereas HR is much slower and takes dozens of hours to complete [69]. In the case of HSGc-C5, the fast-repair process was very fast (12.7 min). This fact indicates, even when we consider the lethality of residual simple DSBs, the number of residual lethal lesion can be very small, because the number of lethal lesions by simple DSBs was calculated by time-integration. Therefore, if the most of the simple lesions are repaired quickly, the number of lethal lesions must be small. Hence, the simple DSBs should not have much of an effect on the SF calculations, as we ignored in this study. This insensitivity of the fast-repair component is briefly discussed in the Appendix B. HR is regarded as one of the most accurate repair processes. According to the estimated lethality, only 3% of repaired lesions through the HR process lead to reproductive cell death of HSGc-C5 cells. This finding indicates that, assuming of that all misrepaired lesions lead to cell death, HR can repair ∼97% of complex DSBs. However, repairing binary lesions is rather difficult. Even if the cell repaired a binary lesions, 40% of rejoined DNA can be misrepaired and leads to cell death.

## 5. Conclusions

To evaluate the repair performance of HSGc-C5 cell against radiation induced DNA damage, the Geant4-DNA application was extended by using newly measured experimental data acquired in this study. Concerning fast- and slow-DNA rejoining, the TLK model parameters were adequately optimized (the repair speeds of each process were reasonably close to the DNA rejoining speed of the NHEJ and HR processes). The lethality rates of the DNA damage induced by complex DSBs and binary repair were approximately 3% and 40%, respectively. Using the optimized repair parameters, the Geant4-DNA simulation was able to predict the SF and the DNA repair kinetics.

## Figures and Tables

**Figure 1 cancers-13-06046-f001:**
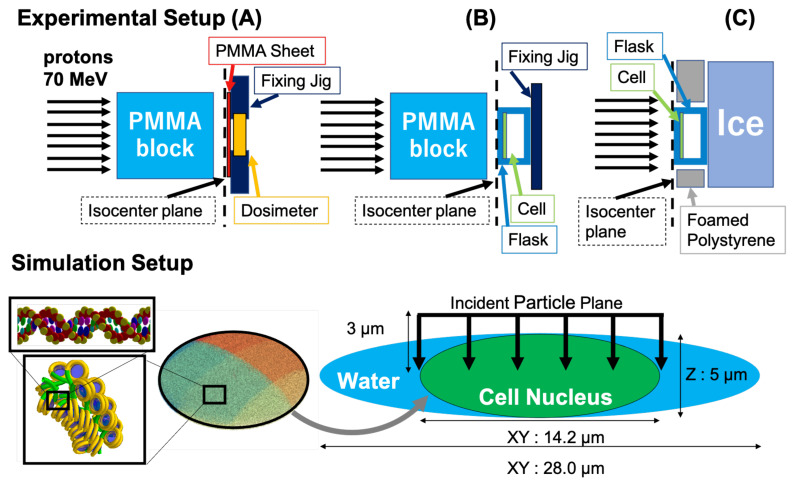
Top (**A**): Schematic experimental setup for dosimetry. Top (**B**): Schematic experimental setup for colony assay. Top (**C**): Schematic experimental setup for the fraction of electrophoresis assay. Bottom: Schematic simulation setup of the irradiated cells and subcomponents.

**Figure 2 cancers-13-06046-f002:**
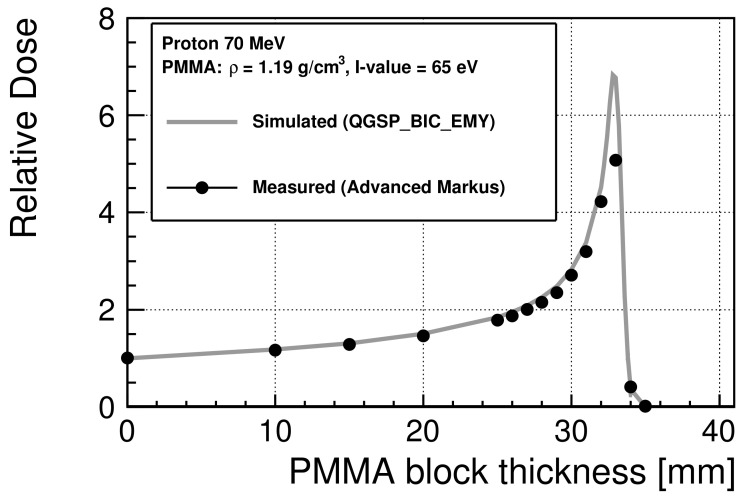
Relative dose of 70 MeV protons in PMMA scaled at 0 mm. The statistical uncertainties are smaller than the marker size. The measured data are available in Appendix A.

**Figure 3 cancers-13-06046-f003:**
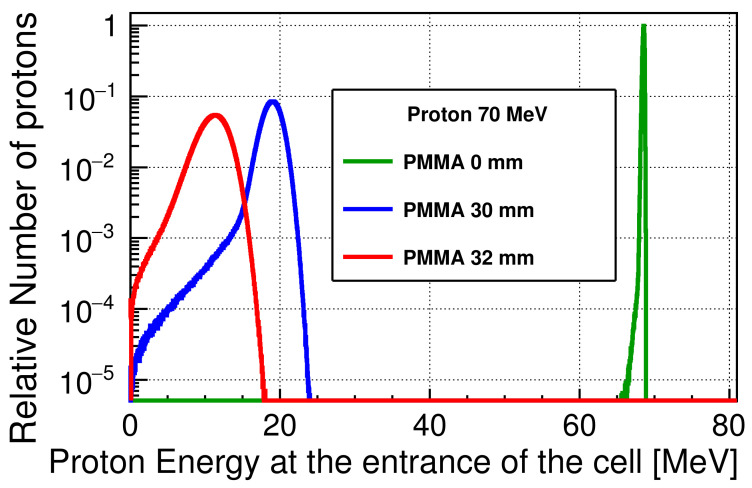
Scaled energy spectra of the 70 MeV incident protons at the cell entrance, downscaled by a PMMA block and a PMMA sheet. The thickness of the PMMA sheet is not considered in the figure legend.

**Figure 4 cancers-13-06046-f004:**
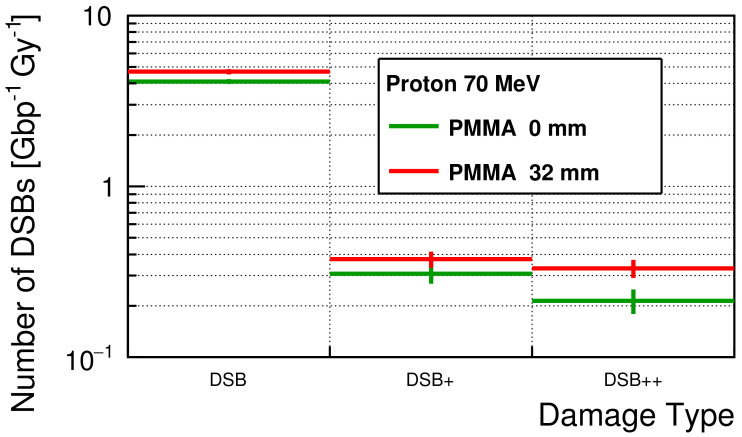
The simulated number of DSBs for each damage type.

**Figure 5 cancers-13-06046-f005:**
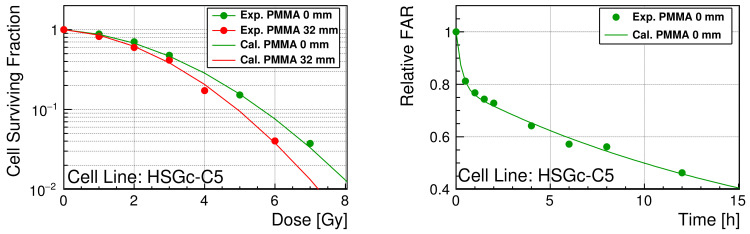
(**Left**): SF of HSGc-C5 as a function of delivered dose. The statistical errors were smaller than the marker size (up to 7%). (**Right**): Relative FAR of HSGc-C5 as a function of time after irradiation. The curve is calculated with the optimized TLK model parameters from the simulated initial DNA damage. The measured SF and relative FAR are available in Appendix A.

**Table 1 cancers-13-06046-t001:** Optimized repair parameters for HSGc-C5.

$\lambda_{1}$ (h${}^{-1}$)	$\lambda_{2}$ (h${}^{-1}$)	η (h${}^{-1}$)	$\beta_{1}$	$\beta_{2}$	γ
3.36	$0.99\times 10^{-2}$	$4.58\times 10^{-6}$	0.	$2.75\times 10^{-2}$	0.39

## Data Availability

All measured data are provided as the Appendix A.

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
