# Peer review of "Performance Evaluation for Repair of HSGc-C5 Carcinoma Cell Using Geant4-DNA"

_cancers, 2021, doi:10.3390/cancers13236046_

Round 1

Reviewer 1 Report

A major revision of the manuscript is needed. The work performed is on improving the Geant4-DNA Monte Carlo simulation using an improved version of the two-lesion kinetic (TLK) model that was recently successfully used to model DNA damages of V79 cells. The authors have re-adjusted the TLK model parameters for a better predictive power of double strand breaks and complex DSBs, including the survival fraction and DNA repair kinetics. However, besides some major English language issues, the authors did not provide a clear presentation of the experimental setup (e.g., schematic in Figure 1 needs to be redone, the referee was confused in the PMMA block vs. sheet usage throughout the text, etc.), presentation of the results (e.g., quantification [chi-squared, log-likelihood or other method], error analysis, etc.), and the discussion of the results. The referee did not perform a comprehensive review of the conclusion, appendix and references since a major revisions of the body must first be addressed. The authors should have a more thorough English language review of the manuscript before re-submission and re-structure the document for better reading clarity.

(Please see attached pdf.file)

Reviewer 2 Report

This is a combined experimental-simulation study using radiobiological cell survival (and DNA damage) data along with the Geant4-DNA simulation toolkit. The work further develops the capacity of Geant4-DNA to contribute to radiobiological research at the subcellular and DNA scale with emphasis on hadron therapeutic issues related to RBE. 

Comments: 

  • Line 38: The papers by Nikjoo et al.  Meas. 41, 1052-1074 (2006), Chatzipapas Med Phys 46, 405 (2019), Chatzipapas et al. Cancers 12, 799 (2020) and Chatzipapas et al. Med Phys 48, 2624 (2021) should be cited for a review of the different MC track structure codes that have been developed for radiobiological research.
  • Line 201: The paper by Kyriakou et al. Medical Physics 42, 3870-3876 (2015) should also be cited in relation to the Geant4-DNA Option 4 models. According to this reference, Option 4 has a limited energy range up to 10 keV. Is this sufficient for the delta rays generated by the incident proton beam?
  • Lines 204-205: Explain what you mean by “proportional probability direct damage model”? Why the PARTRAC (instead of the Nikjoo model used in earlier work) was selected in this regard?
  • Line 206: Is the value of 0.405 determined experimentally or it is an adjustable parameter in the model.
  • Lines 208-209: Can you provide a reference for the “perfect scavenger” assumption used in this study?
  • Lines 218-220: The definitions are not clear. For example, are two DSB further than 10 bp but within the 100 bp limit considered a DSB++? Also, how do you treat two DSB+ within the 100 bp, are they independent?
  • Line 242: If a DSB++ is considered an individual lesion (i.e. the two DSB act as a single lesion which is either repaired or not) then why you multiply by 2.
  • Equation 6: Recast in terms of the time-integrated parameters, as defined in the preceding sentence.
  • Equation 7: The meaning of the sentence that precedes Eq. 7 is unclear.
  • Liners 278: How does the estimated I-value compare with the reference PMMA I-value of ICRU.
  • Line 291: Are these LET values calculated as weighted average from the proton spectra of Fig. 3?
  • Line 295-296: Can you provide an explanation for this observation? Is this simply due an LET effect
  • Figure 4: Although it is hard to find literature data corresponding to identical proton spectra, you may discuss how your DNA damage data compare to literature values for similar proton LET (from other MC studies).
  • Figure 5: What are the standard errors of the data?
  • Line 328: delete “is”
